# The Expression Profile and Prognostic Significance of Metallothionein Genes in Colorectal Cancer

**DOI:** 10.3390/ijms20163849

**Published:** 2019-08-07

**Authors:** Kuo-Chen Hung, Tsui-Chin Huang, Chia-Hsiung Cheng, Ya-Wen Cheng, Ding-Yen Lin, Jhen-Jia Fan, Kuen-Haur Lee

**Affiliations:** 1Division of Gastroenterologic Surgery, Department of Surgery, Yuan’s General Hospital, Kaohsiung 80249, Taiwan; 2Ph.D. Program for Cancer Molecular Biology and Drug Discovery, College of Medical Science and Technology, Taipei Medical University, Taipei 11031, Taiwan; 3Graduate Institute of Cancer Biology and Drug Discovery, College of Medical Science and Technology, Taipei Medical University, Taipei 11031, Taiwan; 4Department of Biochemistry and Molecular Cell Biology, School of Medicine, College of Medicine, Taipei Medical University, Taipei 11031, Taiwan; 5Cancer Center, Taipei Medical University Hospital, Taipei Medical University, Taipei 11031, Taiwan; 6Translational Cancer Research Center, Taipei Medical University, Taipei 11031, Taiwan; 7Department of R&D, Calgent Biotechnology Co., Ltd., Taipei 10675, Taiwan; 8Department of Biotechnology and Bioindustry Sciences, College of Bioscience and Biotechnology, National Cheng Kung University, Tainan 003107, Taiwan; 9Institute of Biochemical Sciences, National Taiwan University, Taipei 10617, Taiwan; 10Food and Drug Administration, Ministry of Health and Welfare, Taipei 11561, Taiwan; 11Cancer Center, Wan Fang Hospital, Taipei Medical University, Taipei 11696, Taiwan

**Keywords:** colorectal cancer, metallothionein, gene signature, prognosis

## Abstract

Colorectal cancer (CRC) is a heterogeneous disease resulting from the combined influence of many genetic factors. This complexity has caused the molecular characterization of CRC to remain uncharacterized, with a lack of clear gene markers associated with CRC and the prognosis of this disease. Thus, highly sensitive tumor markers for the detection of CRC are the most essential determinants of survival. In this study, we examined the simultaneous downregulation of the mRNA levels of six metallothionein (MT) genes in CRC cell lines and public CRC datasets for the first time. In addition, we detected downregulation of these six MT mRNAs’ levels in 30 pairs of tumor (T) and adjacent non-tumor (N) CRC specimens. In order to understand the potential prognostic relevance of these six MT genes and CRC, we presented a four-gene signature to evaluate the prognosis of CRC patients. Further discovery suggested that the four-gene signature (*MT1F*, *MT1G*, *MT1L*, and *MT1X*) predicted survival better than any combination of two-, three-, four-, five-, or six-gene models. In conclusion, this study is the first to report that simultaneous downregulation of six MT mRNAs’ levels in CRC patients, and their aberrant expression together, accurately predicted CRC patients’ outcomes.

## 1. Introduction

Colorectal cancer (CRC) is the third most frequent tumor-related cause of mortality of men and women worldwide [1]. The number of CRC cases is still increasing, and the global burden of CRC is expected to increase by 60% to more than 2.2 million new cases and 1.1 million deaths by 2030 [2]. CRC is a heterogeneous disease; from one patient to another, it differs in clinical presentation, molecular characteristics, and prognosis [3]. Despite comprehensive studies (as reviewed by [4]), the molecular mechanisms of CRC pathogenesis are only partially understood. The formation of CRC involves multiple pathways and stepwise genetic changes; several biomarkers (*APC*, *p53*, *KRAS*, and *BRAF*) are used to detect CRC, but these biomarkers are not sufficiently sensitive and specific. Thus, there is an urgent need for the identification of efficacious biomarkers, therapeutic targets, and agents for early diagnosis, prevention, and personalized therapy in CRC [5]. 

With the development of high-throughput technology, gene expression profiles have been broadly used to identify more novel biomarkers [6]. However, cancer is a heterogeneous disease, resulting from the combined influence of many genetic factors [7]. Therefore, multi-gene biomarkers may be better suited to capturing the complex effects of heterogeneous diseases such as cancer on mRNA abundance levels [8]. Our recent study showed that six metallothionein (MT) genes were among the top 20 downregulated genes in CRC clinical tissues compared with normal colorectal tissues by analysis of a Gene Expression Omnibus (GEO) dataset (GSE21815) (our unpublished data from [9]). MTs are metal-binding proteins involved in diverse processes, including metal homeostasis and detoxification, the oxidative stress response, and cell proliferation [10]. In humans, four MT isoforms exist, labeled by numbers (MT1–MT4). MT2, MT3, and MT4 proteins are encoded by a single gene [11]. MT1 comprises many subtypes encoded by a set of 13 MT1 genes. The known active MT1 genes are *MT1A*, *-B*, *-E*, *-F*, *-G*, *-H*, *-M*, and *-X*. The rest of the MT1 genes (*MT1C*, *-D*, *-I*, *-J*, and *-L*) are pseudogenes whose protein product has not been found in humans [12]. Numerous immunohistochemical and gene expression studies have demonstrated that changes in MT expression are associated with the process of carcinogenesis in various types of human malignancies, including CRC [13,14]. Although downregulation of MT expression has been revealed in association with CRC progression, the prognostic relevance of the expression levels of MTs in CRC is still unclear.

Thus, in the present study, simultaneous downregulation of six MT mRNAs’ levels was examined in CRC cell lines and public CRC datasets. In addition, we also found downregulation of the six MT mRNAs’ levels in 30 pairs of tumor (T) and adjacent non-tumor (N) CRC specimens. Moreover, we further investigated the prognostic significance of these factors for CRC outcomes.

## 2. Results 

### 2.1. mRNA Expression of Six Metallothionein (MT) Genes in Various Human Cancers

Our recent study showed that the metallothionein (MT) gene was one of the most significantly downregulated genes in CRC clinical tissues compared with normal colorectal tissues by analysis of a Gene Expression Omnibus (GEO) dataset (GSE21815) (our unpublished data from reference [9]) (Appendix A). We found that levels of six MT genes were simultaneously decreased from 0.15- to 0.22-fold in CRC tissues compared with in normal colorectal tissues. Next, to further understand the level of MT gene expression in human cancers, we used the National Cancer Institute (NCI)-60 transcriptome database and CellMiner tools [15] (http://discover.nci.nih.gov/cellminer) to determine the mRNA expression of MTs in NCI-60. Figure 1A shows a z-score representation [15] summarizing the various microarray platforms in the NCI database. Among the 60 cell lines, MT transcript levels exhibited low expression in over half of the cell lines. Notably, a single type of cancer cell line, colorectal cancer (CRC), showed markedly low MT transcripts in most CRC cell lines, except for the HCT116 cell line (Figure 1A). Further, to understand the expression levels of MT genes in clinical tissues, we performed Oncomine [16] analysis to investigate differences in the mRNA levels of MTs between tumor and normal tissues in various cancers. As shown in Figure 1B, there were totals of 160, 347, 355, 354, 84, and 340 unique analyses for *MT1B*, *MT1F*, *MT1G*, *MT1H*, *MT1L*, and *MT1X*, respectively. In most of the datasets, mRNA levels of the six MT genes were decreased in most of the tumors, as opposed to in normal tissues. The most notable among these tumors was CRC, which showed the greatest number of cases of decreased expression levels of MT genes. In CRC cases, decreased expression levels of MT genes were observed in a total of 11 datasets for *MT1B*, 24 datasets for *MT1F*, 26 datasets for *MT1G*, 28 datasets for *MT1H*, 9 datasets for *MT1L*, and 25 datasets for *MT1X*. Taken together, these results indicate that mRNA expression levels of MT genes are downregulated in various cancers, especially in CRC.

### 2.2. mRNA Expression of Six MT Genes in Pairs of Tumor (T) and Adjacent Non-Tumor (N) CRC Tissues

The above results indicated that transcript levels of MT genes were significantly downregulated in CRC. Next, resected tumor and corresponding adjacent non-tumor tissues of 30 patients with colorectal cancer were analyzed for MT-mRNA expression. Quantitative RT-PCR analysis was then performed to quantitatively measure the mRNA amount of MT genes in 30 pairs of tumor (T) and adjacent non-tumor (N) CRC specimens. The results showed that *MT1B* was downregulated in about 87% (26/30) of CRC tumor tissues (Figure 2A); *MT1F* was downregulated in about 87% (26/30) of CRC tumor tissues (Figure 2B); *MT1G* was downregulated in about 70% (21/30) of CRC tumor tissues (Figure 2C); *MT1H* was downregulated in about 77% (23/30) of CRC tumor tissues (Figure 2D); *MT1L* was downregulated in about 90% (27/30) of CRC tumor tissues (Figure 2E); and *MT1X* was downregulated in about 80% (24/30) of CRC tumor tissues (Figure 2F). Taken together, these results from clinical specimens indicate that the expression of MTs is significantly downregulated in CRC tumor tissues.

### 2.3. mRNA Expressionof Six MT Genes in Colorectal Cancer Tissues

Further, to confirm the expression levels of the six MT genes in a large number of CRC tissues, we analyzed mRNA expression profiles of six MT genes using existing complementary DNA (cDNA) microarray datasets deposited in the Oncomine database. In the TCGA microarray dataset of the Oncomine database with colorectal tumor and normal colorectal tissues, significant decreases were found in the mRNA expression of *MT1B* (a fold change of −7.717) (Figure 3A), *MT1F* (a fold change of −6.946) (Figure 3B), *MT1G* (a fold change of −10.010) (Figure 3C), *MT1H* (a fold change of −9.898) (Figure 3D), *MT1L* (a fold change of −9.108) (Figure 3E), and *MT1X* (a fold change of −5.364) (Figure 3F) in CRC tissues (*n* = 101) compared to normal colon tissues (*n* = 19). These results suggest that MT genes are downregulated in CRC and may be attractive markers for the diagnosis of CRC.

### 2.4. Prognostic Relevance of the Six Investigated MT Genes in Colorectal Cancer Tissues

We next explored the prognostic relevance of the six MT genes in CRC using *SurvExpress* survival analysis [17]. The patients from the TCGA-CRC dataset (*n* = 350) were classified into predicted low- and high-risk groups according to the prognostic index (PI) (Appendix A). The clinicopathological parameters for the 350 patients involved in this study are supplied in the Appendix A. The results demonstrated that high expression levels of *MT1B*, *MT1F*, *MT1G*, *MT1H*, *MT1L,* and *MT1X* correlated with a low risk (Appendix A). Survival differences between the predicted low- and high-risk groups were evaluated with Kaplan–Meier survival curves and *p* < 0.05 was considered to be statistically significant. There was a significant difference in expression levels of *MT1B* (Figure 4A), *MT1H* (Figure 4D), and *MT1L* (Figure 4E) according to the Kaplan–Meier survival analysis. However, the expression levels of *MT1F* (Figure 4B), *MT1G* (Figure 4C), and *MT1X* (Figure 4F) mRNA were not significantly associated with the clinical outcomes of CRC patients. Collectively, these results indicate that high expression of *MT1B, MT1H,* or *MT1L* correlates with good prognosis in colorectal cancer patients.

### 2.5. A Combination Four-Gene Signature Predicts Survival in Colorectal Cancer Patients

We identified altered expression of the above-mentioned genes to be associated with the prognosis of CRC patients. However, cancer is a heterogeneous disease, and the alternation of a single gene is not sufficient to establish an association between gene and cancer. Thus, multi-gene-combination prediction can improve the sensitivity to the clinical outcomes of cancer patients [18]. Thus, combinations of two-, three-, four-, five-, and six-gene models of CRC patients were analyzed using Kaplan–Meier survival analysis. Specifically, as shown in Appendix A, significant differences in genes selected as a combination of any two-, three-, four-, five-, or six-gene models in clinical outcomes were exhibited according to the Kaplan–Meier survival analysis; in particular, the most significant model was the *MT1F*, *MT1G*, *MT1L*, and *MT1X*-four-gene signature combination. In our four-gene signature, the prognostic index (PI) of the 350 patients was from −9.881 to −7.151, with an optimal cut-off value of −8.134. Those with PI less than −8.134 were placed into the low-risk group (*n* = 195), while those with PI higher than −8.134 formed the high-risk group (*n* = 155). The analysis demonstrated that low risk was correlated with high expression of *MT1F, MT1G, MT1L,* and *MT1X*, while high risk was correlated with low expression of *MT1F, MT1G, MT1L*, and *MT1X* (Figure 5A). In addition, we detected the gene expression levels of *MT1F, MT1G, MT1L,* and *MT1X* in the low-risk and high-risk groups. Our results showed that the gene expression levels of *MT1F*, *MT1G*, *MT1L*, and *MT1X* were higher in the low-risk group than in the high-risk group, and all genes in the four-gene signature showed significant differences (*p* = 7.30 × 10^−3^ for *MT1F*, *p* = 1.20 × 10^−^^2^ for *MT1G*, *p* = 1.97 × 10^−3^^8^ for *MT1L*, and *p* = 4.14 × 10^−^^7^ for *MT1X*) (Figure 5B). Moreover, Kaplan–Meier survival curves showed that patients with a predicted low risk (*n* = 195) had significantly longer survival times than did those with a predicted high risk (*n* = 155) (*p* = 0.00351) (Figure 5C). Taken together, these results suggest that the most significant model of this four-gene signature is related to survival and is a predictor of prognosis in CRC. This may have significant clinical implications for predicting the prognosis of CRC.

## 3. Discussion 

CRC is a heterogeneous disease composed of biologically and clinically diverse diseases. This complexity causes the molecular characterization of CRC to remain deficient, with a lack of clear gene markers associated with CRC and to the prognosis of this disease [19,20]. Thus, highly sensitive tumor markers for the detection of CRC are the most essential determinants of survival. In this study, we identified for the first time that the *MT1F*, *MT1G*, *MT1L*, and *MT1X*-four-gene signature combination is related to survival and is a predictor of prognosis in CRC patients. There are several lines of evidence that support this conclusion. First, we demonstrated simultaneous downregulation of the mRNA levels of six MT genes in CRC cell lines and public CRC datasets. Second, downregulation of the six MT mRNAs’ levels was detected in clinical NT pairs of CRC specimens. Third, we found that high expression of *MT1B*, *MT1H*, or *MT1L* was significantly correlated with good prognosis in CRC patients. Fourth, the most significant four-gene signature model was shown to be related to survival and a predictor of prognosis in CRC. Collectively, this study is the first to report simultaneous downregulation of six MT mRNAs’ levels in CRC patients and their aberrant expression together, accurately predicting CRC patients’ outcomes.

Current molecular changes in colorectal tumors are usually linked to the traditional determination of somatic mutations in well-known tumor-suppressor genes or oncogenes, such as *p53*, *KRAS*, and *BRAF* [21]. Molecular prognosis of CRC tumor samples by transcriptional profiling started about 10 years ago (review in [22]). Despite these efforts, at present, there is not a clear compendium of gene markers for CRC survival, and it is quite difficult to find consistency in the literature [23]. In this study, we identified a group of MT genes as biomarkers, which were downregulated in CRC tumor samples in different public datasets and CRC clinical tissues. In the beginning, six MT genes—*MT1B*, *MT1F*, *MT1G*, *MT1H*, *MT1L*, and *MT1X*—were analyzed among the top 20 downregulated genes in CRC clinical tissues when compared with normal colorectal tissues by analysis GEO dataset (GSE21815) [24]. Further, we found simultaneous downregulation of the six MT mRNAs’ levels in NT pairs of CRC clinical tissues. Moreover, downregulation of the six MT mRNAs’ levels was detected in the TCGA-CRC dataset. We combined different public datasets and CRC clinical tissues to identify six MT genes that had a clear change in expression in CRC tissues and were consistent markers of patient-risk and disease-outcome.

The recent advances in genomic and transcriptomic technologies applied to the study of clinical samples have opened the way to obtaining genome-wide expression profiles of multiple patient cohorts and correlating the expression of certain genes with disease outcome [25,26]. Importantly, some prognostic models based on gene expression levels are an excellent tool to investigate the prognosis of disease and to build risk predictors that will be applicable to individual patients. In our study, we analyzed the association of *MT1B*, *MT1F*, *MT1G*, *MT1H*, *MT1L*, and *MT1X* single gene expression with the prognosis of CRC patients in the TCGA-CRC dataset from the *SurvExpress* database. The data demonstrated that low expression of *MT1B, MT1H,* or *MT1L* was significantly correlated with a high risk of poor prognosis (*p* < 0.05). However, the efficacy of a single index was limited. Therefore, multi-gene-combination prediction can improve the sensitivity to clinical outcomes of heterogeneous diseases such as cancer to mRNA abundance levels. Thus, we identified for the first time, the most significant four-gene signature model (*MT1F*, *MT1G*, *MT1L*, and *MT1X*) that was able to predict survival and CRC prognosis. Overall, this multi-gene panel may serve as a promising outcome predictor and as potential therapeutic targets in CRC patients.

In conclusion, we consider that the results presented in this work provide strong support and a solid rationale for the exploration of changes in expression of MTs in CRC to assist in the development of clinically useful outcome prediction of CRC. 

## 4. Materials and Methods

### 4.1. Tissue Samples and Ethics Statement

Human 30 pairs of tumor (T) and adjacent non-tumor (N) CRC specimens were obtained from the Department of Surgery, Taipei Medical University Hospital (Taipei, Taiwan). Informed written consent was obtained from all patients and/or guardians for the use of their resected specimens. Acquisition of samples and their subsequent examination were approved by the Institutional Review Board (IRB) of Taipei Medical University (TMU-JIRB No.: 201312039). None of the participants had a previous history of cancer.

### 4.2. Quantitative Reverse-Transcription (RT)-(q)PCR

Total RNA was extracted from NT pairs of CRC tissues using a Qiagen RNeasy kit (Qiagen, Valencia, CA, USA) and Qiashredder columns according to the manufacturer’s instructions (Qiagen). One microgram of total RNA was reverse-transcribed to complementary (c)DNA using a Reaction Ready™ First Strand cDNA Synthesis Kit (SABiosciences, Frederick, MD, USA). To detect mRNA expression levels of MT genes and GAPDH, specific products were amplified and detected using the cycle profile of the Qiagen SYBR green reagent (Qiagen). The sequences of primers used in this study are listed as follows: MT1B: forward 5′-GCACCACAGGTGGCTCCTG-3′ and reverse 5′-TGGGCACACTTGGCACAGC-3′; MT1F: forward 5′-CAACTGCTCCTGCGCCGCT-3′ and reverse 5′-CCAGACCATGGAGATGGCC-3′; MT1G: forward 5′-CGCTGCAGGTGTCTCCTGC-3′ and reverse 5′-CAGCCCTGGGCACACTTG-3′; MT1H: forward 5′-CCTGCGAGGCTGGTGGCTC-3′ and reverse 5′-CTGGGCACACTTGGCACAG-3′; MT1L forward 5′-GCTCCTGCTCCTGTGCCAG-3′ and reverse 5′-TCTGGGAGCAGGGCTGTCC-3′; MT1X: forward 5′-GCTCGCCTGTTGGCTCCTG-3′ and reverse 5′-TGCAGATGCAGCCCTGGGC-3′; and GAPDH: forward 5′-AATCCCATCACCATCTTCCA-3′ and reverse 5′-TGAGTACGTCGTGGAGTCCA-3′.

### 4.3. CellMiner Data Mining and Analysis

The CellMiner tool (http://discover.nci.nih.gov/cellminer; version 1.5) was used to compare and plot the relative baseline expression of MT mRNA in the NCI-60 cell line panel. The tool enables retrieval and integrated analysis of baseline and experimental data compiled from the 60 cell lines included in the panel. CellMiner gene transcript data were generated from microarray platforms. We selected gene transcript level z-scores for analysis of the six MT genes as gene identifier inputs.

### 4.4. Oncomine Database Analysis

Gene expression changes were analyzed in the TCGA microarray dataset of the Oncomine website with colorectal tumor and normal colorectal tissues (www.oncomine.org, Compendia biosciences, Ann Arbor, MI, USA). The threshold search criteria used in the study were a *p*-value of <0.001, a fold change of >2, and a gene rank in the top 5%.

### 4.5. SurvExpress Database Analysis

In our analysis, *SurvExpress* was used to provide survival analysis and risk assessment. *SurvExpress* (http://bioinformatica.mty.itesm.mx/SurvExpress), which is a comprehensive gene expression database, can provide risk assessment and survival analysis in cancer datasets using a biomarker gene list as an input. The samples of each dataset were split into two risk groups with the same size; each group was determined according to the ordered prognostic index (PI) with the dataset split by the ordered PI (higher values for lower risk) so as to have an equal number of samples in each group. The PI was computed using the expression levels and values obtained from the Cox fitting algorithm [27].

### 4.6. Statistical Analysis

*p*-Values and fold-changes for differential expression analysis of genes generated from NT pairs of CRC tissues and the Oncomine database were calculated using a one-sided Student’s *t*-test. *p* values of <0.05 were considered significant.

## Figures and Tables

**Figure 1 ijms-20-03849-f001:**
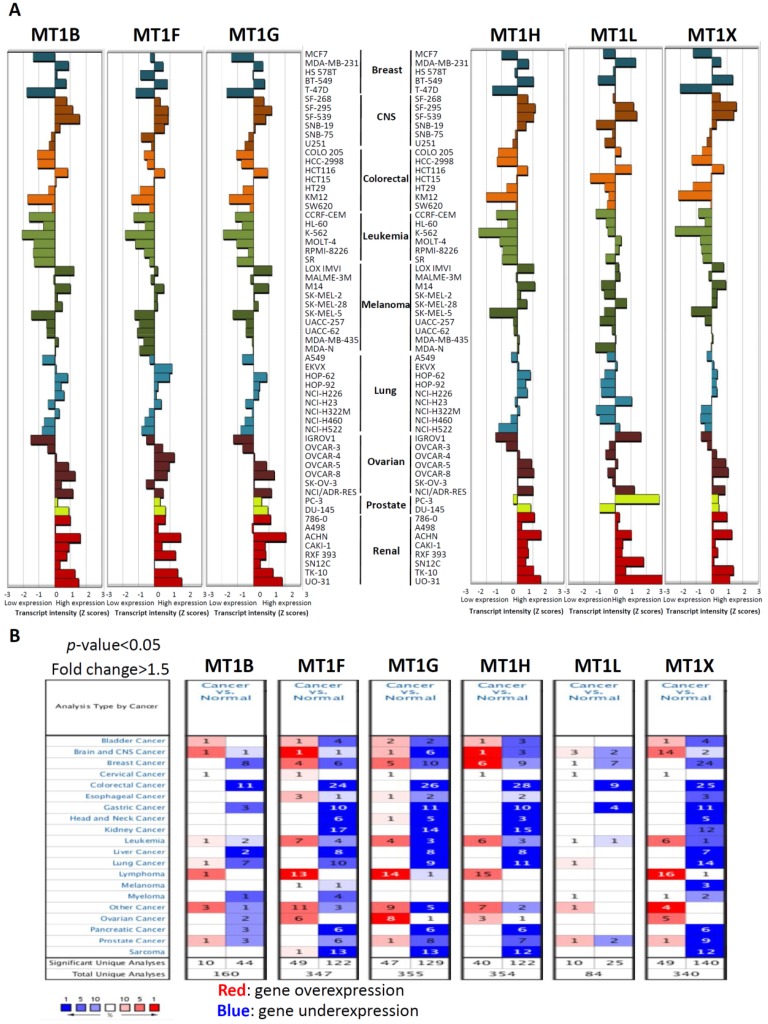
Metallothionein (MT) mRNA expression in the National Cancer Institute (NCI)-60 human tumor cell lines and various cancer tissues. (**A**) Relative MT gene expression profile. Bars to the right show high expression, while bars to the left show low expression relative to the expression mean. Expression values are normalized as z-scores. Data are accessible at http://discover.nci.nih.gov/cellminer. (**B**) Expressions of MT mRNA in 20 common cancers were compared with those in corresponding normal tissues (Oncomine Database). The search criteria thresholds for datasets of cancer versus normal analysis were a *p*-value of <0.05, a fold change of >1.5, and a gene rank in the top 10%. Red signifies gene overexpression in the analyses; blue represents gene underexpression.

**Figure 2 ijms-20-03849-f002:**
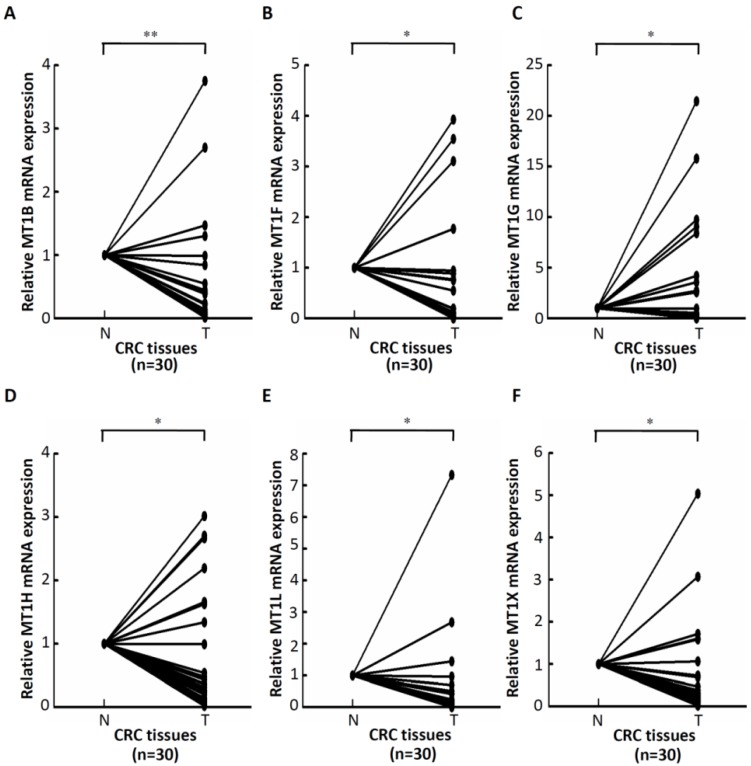
The mRNA expression of six MT genes in 30 pairs of tumor (T) and adjacent non-tumor (N) CRC specimens. The 30 NT paired tissues were used for examination of the expression of the six MT genes. Lower expression of *MT1B* (**A**), *MT1F* (**B**), *MT1G* (**C**), *MT1H* (**D**), *MT1L* (**E**), and *MT1X* (**F**) was found in tumor part specimens than in non-tumor part tissues. * *p* < 0.05, ** *p* < 0.01.

**Figure 3 ijms-20-03849-f003:**
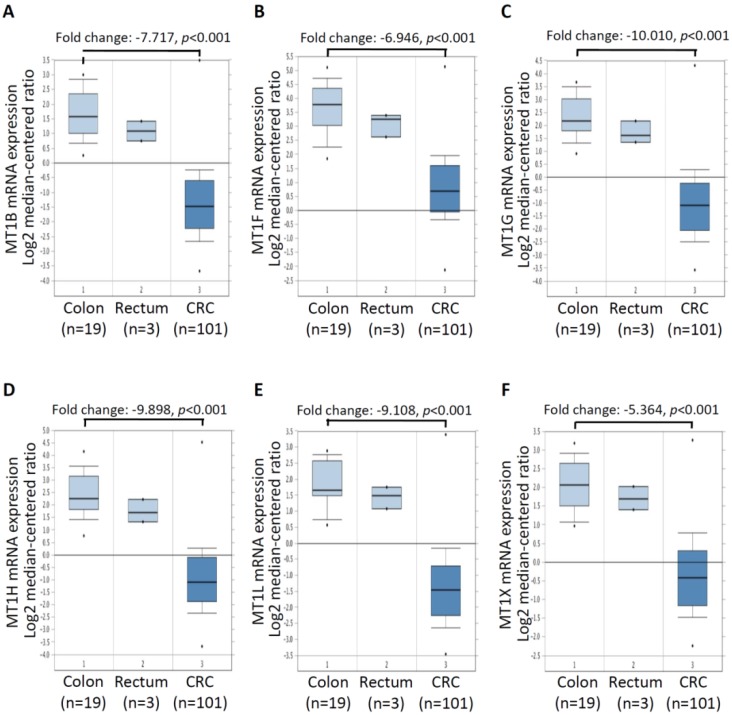
MT genes were downregulated in CRC tissues. Relative expression levels of *MT1B* (**A**), *MT1F* (**B**), *MT1G* (**C**), *MT1H* (**D**), *MT1L* (**E**), and *MT1X* (**F**) in colon/rectum normal tissue and CRC tissues by using the Oncomine database (http://www.oncomine.org/).

**Figure 4 ijms-20-03849-f004:**
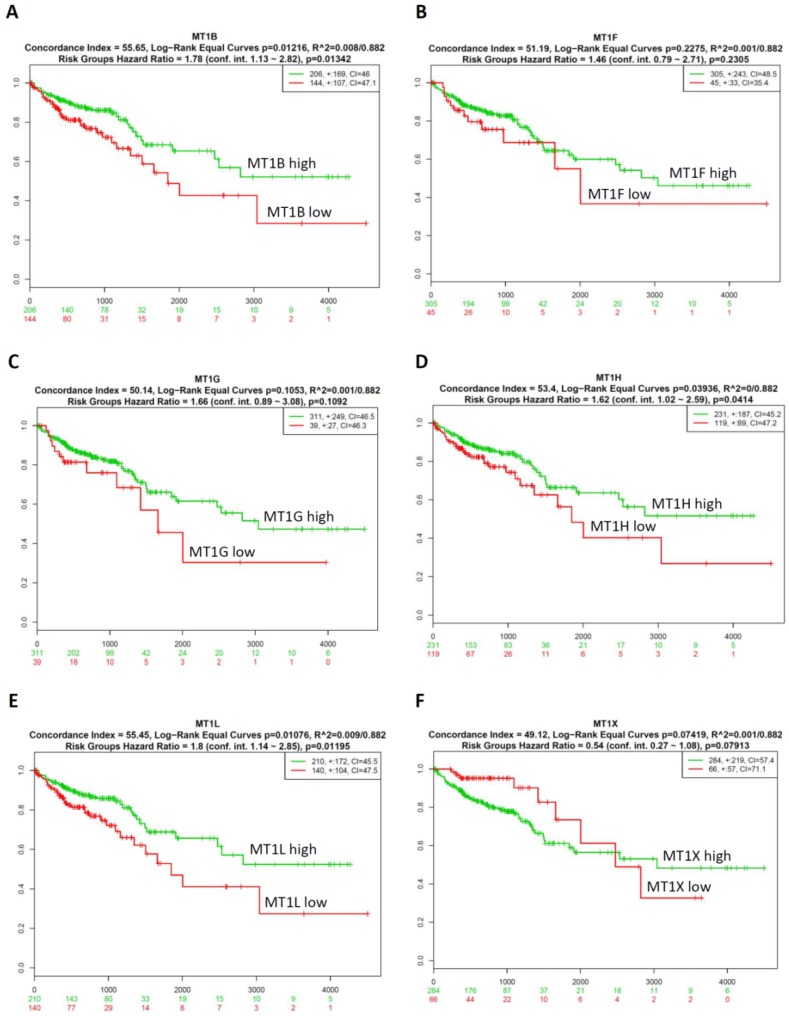
Expression levels of MT genes were associated with prognosis in patients with CRC. Low expression levels of *MT1B* (**A**), *MT1F* (**B**), *MT1G* (**C**), *MT1H* (**D**), *MT1L* (**E**), and *MT1X* (**F**) were correlated with shorter survival time of CRC patients. Green and red lines indicate high- and low-risk groups, respectively. *p* < 0.05 was considered to be statistically significant.

**Figure 5 ijms-20-03849-f005:**
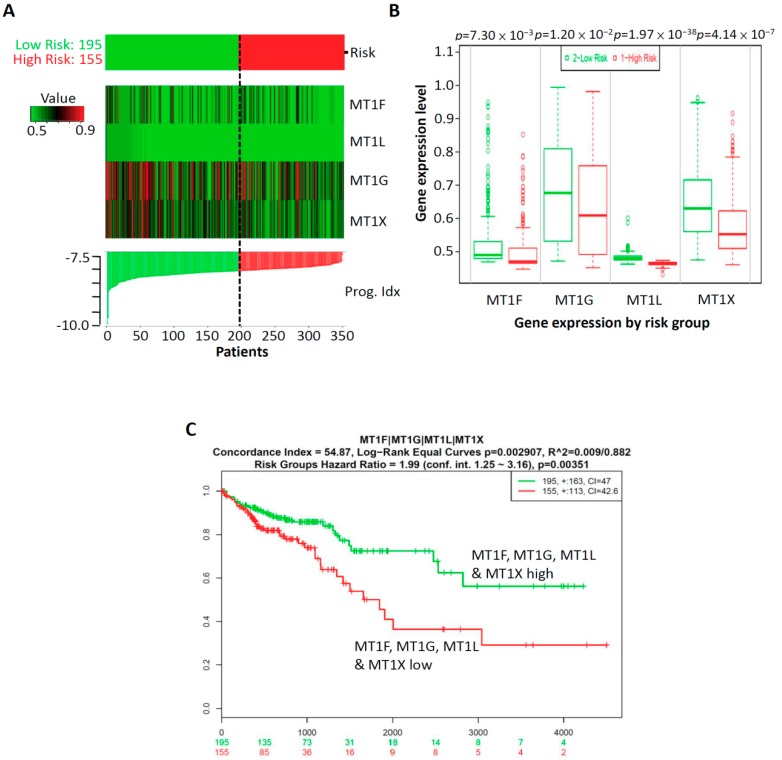
Clinical outcomes for the four-gene combination predicted survival better than the individual genes did alone in CRC patients. (**A**) The *SurvExpress* database was used to analyze the association of the four-gene signature with the predicted risk. (**B**) The gene expression levels of *MT1F*, *MT1G*, *MT1L*, and *MT1X* were detected in high-risk and low-risk groups. (**C**) Kaplan–Meier survival curves showed that patients with high levels of *MT1F*, *MT1G*, *MT1L*, and *MT1X* (*n* = 195) had significantly longer survival times than did those with low levels of *MT1F*, *MT1G*, *MT1L*, and *MT1X* (*n* = 155). *p* < 0.05 was considered to be statistically significant.

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
