# Peer review of "The Expression Profile and Prognostic Significance of Metallothionein Genes in Colorectal Cancer"

_ijms, 2019, doi:10.3390/ijms20163849_

Round 1

Reviewer 1 Report

1. In the title there is a statement concerning "metallothioneins” but the research concern only isoforms MT-1, not MT-2,3 and 4. It should be precise in few places in the text

2. Some phrases are wrote in another type, for example “multifactorial disease”

3. Abbreviation “NT pair” is not explained

4. Two times in the text 9once in the introduction and once in the results) authors said : Our recent study showed that six metallothioneins (MTs) were bellowed to top 20 downregulated genes in CRC clinical tissues to compare with colorectal normal tissues by analysis GEO dataset (GSE21815) (unpublished data) [9] . It should e explained that there are not published but the reference is from Biochim Biophys Acta 2015,

 5. Abbreviation “GEO” is not explained in the first place of used but in the another one

6, Lane 84- authors stated that: “Among 60 cell lines, MTs transcript levels exhibit low  expression in over than half of cell lines.“ What does it mean “over”? From the figure 1, is rather about half

 7. It should be discussed what is the difference between the line HCY116 from the others CRC cell lines. Maybe is worth of knowing and it can explained the difference.

 8. In the all figures “fig….”should be delated.

9. line 169, there is “MT1” without letter

10. Lines 209-211, authors said “we” but the reference is from another group. Should be corrected

Author Response

Manuscript ID: ijms-559101

Title: The Expression Profile and Prognostic Significance of Metallothionein Genes in Colorectal Cancer

Reviewer Comments

Reviewer 1

We appreciate the comments of this reviewer and believe that our manuscript has been improved by attention to him or her. The followings are our responses to the specific issues raised by this reviewer:

Point 1: In the title there is a statement concerning "metallothioneins” but the research concern only isoforms MT-1, not MT-2,3 and 4. It should be precise in few places in the text. 

Response 1: We are sorry for the misleading and thank reviewer's valuable suggestion. We have revised the title as “The Expression Profile and Prognostic Significance of Metallothionein Genes in Colorectal Cancer” in the revised manuscript. In this study, six MT genes—MT1B, MT1F, MT1G, MT1H, MT1L and MT1X—were belong to the metallothionein genes.

Point 2: Some phrases are wrote in another type, for example “multifactorial disease”.

Response 2: We are sorry for the misleading and thank reviewer's valuable suggestion. We have revised the “multifactorial disease” as “heterogeneous disease“ in the revised manuscript.

Point 3: Abbreviation “NT pair” is not explained.

Response 3: We are sorry for the misleading and thank reviewer's valuable suggestion. We have revised “NT pair” as tumor (T) and adjacent non-tumor (N) in the revised manuscript.

Point 4: Two times in the text 9 once in the introduction and once in the results) authors said : Our recent study showed that six metallothioneins (MTs) were bellowed to top 20 downregulated genes in CRC clinical tissues to compare with colorectal normal tissues by analysis GEO dataset (GSE21815) (unpublished data) [9] . It should e explained that there are not published but the reference is from Biochim Biophys Acta 2015,

Response 4: Thanks for reviewer's valuable comments. The sentence has been revised in revised manuscript and shown below:

Our recent study showed that the metallothionein (MT) gene was one of the most significantly downregulated genes in CRC clinical tissues compared with normal colorectal tissues by analysis of a Gene Expression Omnibus (GEO) dataset (GSE21815) (our unpublished data from reference [9]).

Point 5: Abbreviation “GEO” is not explained in the first place of used but in the another one

Response 5: Thanks for reviewer's valuable comments. Abbreviation “GEO” has been explained in the first place in revised manuscript and shown below:

Gene Expression Omnibus (GEO) dataset

Point 6: Lane 84- authors stated that: “Among 60 cell lines, MTs transcript levels exhibit low expression in over than half of cell lines.“ What does it mean “over”? From the figure 1, is rather about half

Response 6: We are sorry for the misleading and thank reviewer’s valuable comments. The sentence has been revised in revised manuscript and shown below:

Among the 60 cell lines, MT transcript levels exhibited low expression in over half of the cell lines.

Point 7: It should be discussed what is the difference between the line HCT116 from the others CRC cell lines. Maybe is worth of knowing and it can explained the difference.

Response 7: Thanks for reviewer's valuable comments. It has been demonstrated that MT gene—MT1F expression was significantly downregulated in colon cancer cell lines (HT29, SW620, LOVO) due to high frequency of loss of heterozygosity (LOH) at MT1F gene locus, except for the HCT116 cell line [1]. This result is consistent with our finding. Thus, high frequency of LOH at MT1F gene locus may be one of possibility to cause low expression of MT1F in the others CRC cell lines.

Reference

[1] Yan, D.W., Fan, J.W., Yu, Z.H., Li, M.X., Wen, Y.G., Li, D.W., Zhou, C.Z., Wang, X.L., Wang, Q., Tang, H.M., Peng, Z.H., Downregulation of metallothionein 1F, a putative oncosuppressor, by loss of heterozygosity in colon cancer tissue. Biochim Biophys Acta. 2012, 1822, 6, 918-26.

Point 8: In the all figures “fig….”should be deleted.

Response 8: We are sorry for the misleading and thank reviewer’s valuable comments. In the all figures “fig….”has been deleted in revised manuscript.

Point 9: line 169, there is “MT1” without letter

Response 9: We are sorry for the misleading and thank reviewer’s valuable comments. We have revised “MT1” as “MT1X” in the revised manuscript.

Point 10: Lines 209-211, authors said “we” but the reference is from another group. Should be corrected

Response 10: We are sorry for the misleading and thank reviewer’s valuable comments. The sentence has been revised in revised manuscript and shown below:

In the beginning, six MT genes—MT1B, MT1F, MT1G, MT1H, MT1L and MT1X—were analyzed among the top 20 downregulated genes in CRC clinical tissues when compared with colorectal normal tissues by analysis GEO dataset (GSE21815) [24].

Reviewer 2 Report

The authors describe an intriguing study that presents a four-gene signature for patient prognosis and reports the simultaneous downregulation of specific mRNA levels in CRC patients and their relevance towards predicting patient outcomes. A few comments:

1.      The resolution of figure 1 is too low. Also, please enlarge the font

2.      Extensive revision of grammar and sentence structure is required e.g. “Figure 4. Expression levesl”, “considered to be statistically significant. .”, “The recent advance of genomic”, “mrna

3.      The result from these overall analyses was interesting but could benefit by breaking down the analyses to particular clinicopathological parameters, such as cancer stage, subtype etc. This could provide more insights to realise the four-gene signature as a panel for diagnosis and prognosis.

4.      Expression levels of MT1F (Figure 4B), MT1G (Figure 4C), or MT1X (Figure 4F) mRNA were not significantly associated with the clinical outcomes of CRC patients. Could the authors provide prior literature or molecular analysis to further discuss the reason and impact of these findings?

5.      A complete listing of the clinicopathological parameters for the 350 patients involved in this study should be listed in the supplementary

6.      The study would benefit greatly if the gene panel was tested on fresh clinical samples, on a single-cell basis, due to tumor heterogeneity

7. The authors can consider looking for gene datasets of colorectal circulating tumor cells to validate their gene panel as well

Author Response

Manuscript ID: ijms-559101

Title: The Expression Profile and Prognostic Significance of Metallothionein Genes in Colorectal Cancer

Reviewer Comments

Reviewer 2

We appreciate the comments of this reviewer and believe that our manuscript has been improved by attention to him or her. The followings are our responses to the specific issues raised by this reviewer:

Point 1: The resolution of figure 1 is too low. Also, please enlarge the font.

Response 1: We would like to thank reviewer for the comments. We have revised the Fig.1 by improving resolution and enlarging the font in the revised manuscript and shown below:

Point 2: Extensive revision of grammar and sentence structure is required e.g. “Figure 4. Expression levesl”, “considered to be statistically significant. .”, “The recent advance of genomic”, “mrna”.

Response 2: Thanks for reviewer's valuable comments. Our manuscript have sent to MDPI for English editing. The English-editing-certificate shows below:

Point 3: The result from these overall analyses was interesting but could benefit by breaking down the analyses to particular clinicopathological parameters, such as cancer stage, subtype etc. This could provide more insights to realise the four-gene signature as a panel for diagnosis and prognosis.

Response 3: Thanks for reviewer's valuable comments. It is important for the establishment the relationship between multiple-gene expression signature and clinicopathological characteristics. But, in our study, the prognostic relevance of the MT1F, MT1G, MT1L, and MT1X-four-gene signature in CRC using SurvExpress survival analysis. SurvExpress is a cancer-wide gene expression database to provide the prediction the multi-gene biomarkers for clinical outcomes. This database does not provide the prediction for multi-gene biomarkers to clinicopathological parameters.

Point 4: Expression levels of MT1F (Figure 4B), MT1G (Figure 4C), or MT1X (Figure 4F) mRNA were not significantly associated with the clinical outcomes of CRC patients. Could the authors provide prior literature or molecular analysis to further discuss the reason and impact of these findings?

Response 4: We would like to thank reviewer for the comments. It has been demonstrated that MT1F mRNA expression was significantly downregulated in colon cancer cells, however, its mRNA level was not associated with the clinicopathological characteristics of the CRC patients [1]. For MT1G and MT1X, there is no report about the mRNA expression level of MT1G or MT1X and clinical outcomes of CRC patients. The molecular analysis for the downregulation MT1F mRNA expression, it has reported that loss of heterozygosity (LOH) at MT1F gene locus to cause low expression of MT1F in CRC cell lines [1].

Reference

[1] Yan, D.W., Fan, J.W., Yu, Z.H., Li, M.X., Wen, Y.G., Li, D.W., Zhou, C.Z., Wang, X.L., Wang, Q., Tang, H.M., Peng, Z.H., Downregulation of metallothionein 1F, a putative oncosuppressor, by loss of heterozygosity in colon cancer tissue. Biochim Biophys Acta. 2012, 1822, 6, 918-26.

Point 5: A complete listing of the clinicopathological parameters for the 350 patients involved in this study should be listed in the supplementary.

Response 5: We thank reviewer’s valuable comments. We have listed the clinicopathological parameters for the 350 patients involved in this study in the supplementary Table 2 of revised manuscript. 

Point 6: The study would benefit greatly if the gene panel was tested on fresh clinical samples, on a single-cell basis, due to tumor heterogeneity

Response 6: Thanks for reviewer's valuable comments. Single-cell basis approach is a powerful novel technique for studying the molecular underpinnings of cellular heterogeneity and consequences of cellular variability. We will keep in mind the reviewer's valuable comments for the further experiment design.

Point 7: The authors can consider looking for gene datasets of colorectal circulating tumor cells to validate their gene panel as well

Response 7: We would like to thank reviewer for the comments. Circulating tumor cells (CTCs) have become a hot topic of discussion for oncologists because of their tremendous potential in the diagnosis and treatment of cancer [1]. In colorectal cancer, CTCs could represent early prognostic and predictive markers in early and metastatic CRC, revealing crucial information on disease monitoring [2]. However, there is no any gene datasets of colorectal circulating tumor cells. We will keep in mind the reviewer's valuable comments for the further experiment design about our four-genes model and colorectal CTCs. 

References

[1] Pravin, D. P., Navjeet, K. L., Role of circulating tumor cells in future diagnosis and therapy of cancer. J Cancer Metastasis Treat 2015, 1, 44–56.

[2] Claudia, B., Vlad-Vasile, P., Rare,s B., Sur, D., Gabriel, S., Cornel, A., Iulia, L., Circulating tumor cells in clinical research and monitoring patients with colorectal cancer. Oncotarget 2018, 9, 24561–24571.